# Vertically Resolved Formation Mechanisms of Fine Particulate Nitrate in Asian Megacities: Integrated Lidar – Aircraft Observations and Process Analysis

Yutong Tian[1,3], Ting Yang[1*], Hongyi Li[1], Ping Tian[2], Yifan Song[1,3], Jiancun He[1,4], Yining Tan[1,3], Yele Sun[1,3], Zifa Wang[1,3]

[1]State Key Laboratory of Atmospheric Environment and Extreme Meteorology, Institute of Atmospheric Physics, Chinese Academy of Sciences, Beijing, 100029, China.

[2]Beijing Weather Modification Office, Beijing, 100089, China.

[3]College of Earth and Planetary Sciences, University of Chinese Academy of Sciences, Beijing, 100049, China.

[4]College of Advanced Agriculture and Ecological Environment, Heilongjiang University, Harbin, 150080, China.

*Correspondence to*: Ting Yang (tingyang@mail.iap.ac.cn)

**Abstract.** The vertical distribution of particulate nitrate is crucial for understanding its formation mechanisms and developing urban haze reduction strategies. Advanced technologies were used in this study to collect continuous vertical data on nitrate concentrations in Beijing for 2021, providing a seasonal analysis of their distribution and influencing factors. Spring exhibited the highest nitrate concentration below 2 km ($8.29 \pm 3.14$ μg/m$^3$), followed by winter ($7.34 \pm 2.78$ μg/m$^3$), autumn ($6.65 \pm 2.11$ μg/m$^3$), and summer ($2.23 \pm 0.82$ μg/m$^3$). Below 300 m, thermodynamic factors dominated nitrate formation in spring and summer (RH: R = 0.64; temperature: R = −0.76), whereas winter formation was driven by both atmospheric oxidizing capacity (AOC, R = 0.52) and thermodynamic factors (R = 0.68). Between 0.8 km and 2 km, dynamic drivers prevailed in spring and autumn (TKE: R = -0.41; vertical wind speed: R = -0.43), while photochemical factors dominated in winter and summer (AOC: R = 0.58; O$_3$: R = 0.60). High nitrate levels were observed at the boundary layer top (0.7–1.2 km), peaking at 118.11 μg/m$^3$ in late autumn, closely linked to photochemical processes and dynamic drivers. In winter, nitrate concentrations exhibited distinct diurnal variations, peaking at 13:00, 18:00, and 22:00, with variations and peak concentrations increasing with altitude due to the accumulation of photochemical products and enhanced AOC at night. These findings support targeted emission controls by reducing photochemical precursor emissions at the boundary layer top and strengthening NO$_x$ reductions at major sources.

## 1 Introduction

Haze represents a significant atmospheric pollution phenomenon, particularly in densely populated and highly industrialized regions, where it has emerged as an environmental issue of considerable concern (Guan et al., 2024; Kong et al., 2024; Zhao et al., 2020). This phenomenon is primarily driven by elevated concentrations of fine particulate matter with diameters less than 2.5 μm (PM$_{2.5}$) and its components in the atmosphere, which not only degrades air quality and visibility but also poses

substantial risks to public health, influences regional climate patterns, and contributes to extreme weather events and meteorological hazards (Bălă et al., 2021; Weichenthal et al., 2024). Secondary inorganic aerosols, especially sulfates ($SO_4^{2-}$), nitrates ($NO_3^-$), and ammonium ($NH_4^+$), which are constitute critical components of $PM_{2.5}$ (Meng et al., 2022; He et al., 2012; Liu et al., 2022). Among these, particulate nitrate is a principal chemical constituent of $PM_{2.5}$, with significant implications for air quality, ecosystems, climate change, and human health (Li et al., 2021; Slawsky et al., 2021). Particulate nitrate facilitates

the hygroscopic growth of particulate matter and alters the optical properties of $PM_{2.5}$ (Liu et al., 2020b; Luo et al., 2021). Notably, nitrate aerosols exhibit a more substantial scattering effect on visible light at lower relative humidity than sulfates, contributing more significantly to haze formation (Li et al., 2022; Li et al., 2018). Furthermore, nitrate aerosols can irritate the human respiratory system, and prolonged exposure to environments with high nitrate concentrations is associated with an increased risk of respiratory diseases (Brender et al., 2011). The escalating issue of particulate nitrate pollution presents a

formidable challenge to ongoing efforts to improve air quality (Zhai et al., 2021; Sun et al., 2013b).

Since the Clean Air Action launched in 2013, the annual average concentration of $PM_{2.5}$ has effectively decreased, but nitrate concentrations have not shown a consistent downward trend and have in fact increased during severe pollution events in winter, particularly in Beijing (Cao et al., 2022). This phenomenon has made nitrate an essential component of winter haze in Beijing (Feng et al., 2021). The recent rise in the proportion of particulate nitrate in $PM_{2.5}$ mass is primarily attributed to the reduction

in sulfur dioxide ($SO_2$) emissions, enhanced atmospheric oxidative capacity, and weakened dry deposition (Fu et al., 2020). The chemical conversion of nitrogen oxides ($NO_x$) to nitrate mainly involves gas-phase oxidation and heterogeneous processes, which exhibit significant diurnal variations: gas-phase reactions dominate nitrate formation during the day, while with weak photochemical reactions in the nighttime, heterogeneous processes become critical pathways for nitrate generation (Xie et al., 2022). During winter pollution periods in China, the formation of particulate nitrate is predominantly driven by gas-phase

oxidation, significantly surpassing the contribution of dinitrogen pentoxide ($N_2O_5$) heterogeneous uptake at nighttime (Chen et al., 2020). Moreover, the contribution of nitrate in the nocturnal residual layer is often overestimated. Although high ozone ($O_3$) concentrations in the residual layer can potentially drive the conversion of $NO_x$ to nitrate, unfavourable meteorological conditions and low specific surface area of particles result in very low nitrate formation efficiency (Tang et al., 2021). Therefore, the nitrate formation processes in the atmosphere are complex and challenging to elaborate.

Changes in nighttime oxidation are also a crucial factor in the increase of nitrate concentrations (Brown and Stutz, 2012). Despite a global decrease in nitrate radicals ($NO_3$) (Archer-Nicholls et al., 2023; Wang et al., 2023a), nighttime oxidation in China has significantly intensified, leading to a shortened nighttime lifetime of $NO_x$, affecting $O_3$ and $PM_{2.5}$ pollution levels. More specifically, nighttime $NO_3$ radicals react with volatile organic compounds (VOCs), promoting the formation of secondary organic aerosols (Wang et al., 2023b). The impact of increased photochemical oxidants on nitrate formation is

emphasized. Winter photochemical reactions remain strong enough to prevent a significant decrease in nitrate formation even with reduced $NO_x$ emissions (Yang et al., 2024).

Methods to explore the vertical distribution of atmospheric particulate chemical components include ground monitoring, aerial surveys, airborne balloons, vertically mobile pods, satellite remote sensing, model simulation, etc (Wang et al., 2022). However,

due to the limitations of the number of ground stations and aerial routes and the lack of understanding of non-homogeneous formation mechanisms of particulate nitrate, it is very difficult to obtain accurate and continuous vertical distribution information of nitrate on a long time scale with a high spatial and temporal resolution (Fan et al., 2022). Recent studies on the vertical distribution of fine particulate matter have employed tower measurement techniques at various heights in locations such as Beijing, Guangzhou and the Pearl River Delta region, which utilized platforms at multiple altitudes, allowing for a comprehensive assessment of how particulate matter varies with height in urban environments (Sun et al., 2015b; Zhou et al., 2020). The findings consistently indicated that nitrate concentrations tend to increase with altitude within the lower boundary layer (0–300 m), suggesting that higher levels experience enhanced nitrate generation (Fan et al., 2021). This phenomenon is attributed to nighttime processes, such as heterogeneous reactions, which are more pronounced at elevated altitudes. Additionally, studies highlighted the significance of particulate partitioning and non-homogeneous phase reactions in contributing to nitrate formation in the upper layers of the urban atmosphere (Zhou et al., 2019; Yang et al., 2021a). Despite these insights, researches are limited by the absence of continuous vertical observations and the temporal discontinuities associated with the observational methods employed (Lin et al., 2022).

In this study, we leveraged a novel approach by utilizing remote sensing retrieval data of $PM_{2.5}$ chemical components for the entire year of 2021 in the urban area of Beijing. This method provided vertically continuous, high temporal resolution, and long-term data on mass concentrations. We divided different altitude layers vertically to comprehensively investigate the seasonal variations in nitrate concentrations and formation mechanisms in urban Beijing, and coefficient analyses of various driving factors were conducted in relation to nitrates. Additionally, we conducted a detailed analysis of specific pollution events. This study enhances the understanding of atmospheric physics and chemistry, providing insights and recommendations for mitigating nitrate pollution across various altitude layers in Beijing throughout the four seasons.

## 2 Data and methods

### 2.1 Components data

#### 2.1.1 Retrieval data

Beijing faces severe pollution challenges due to its unique topography and rapid economic development, making it an ideal location to study issues related to varying pollution levels. This study employed a remote sensing retrieval technique based on LiDAR and sun photometer data (Wang et al., 2022) to estimate six optical variables: inorganic nitrate (AN), aerosol water (AW), water-soluble organic matter (WSOM), insoluble organic matter (WIOM), black carbon (BC) and the extinction coefficient at 532 nm. The LiDAR system is installed on the roof of a 28-meter-tall building at the tower branch of Institute of Atmospheric Physics, Chinese Academy of Sciences (IAP, CAS, 39◦58′35″N, 116 ◦22′41″E).

We developed a deep-learning mapping model to convert optical components into chemical ones by integrating a convolutional neural network (CNN) for spatial-feature extraction with a long short-term memory (LSTM) network for

capturing temporal dependencies (Hinton et al., 2006; Wang et al., 2016). Bayesian optimization (BO) was then applied to automate hyperparameter tuning (Frazier, 2018). The model ingests twelve input variables including six optical variables, geopotential height, relative humidity, temperature as well as the u, v and w wind components. It outputs mass concentrations of $NH_4^+$, $NO_3^-$, $SO_4^{2-}$, OM and BC. The resulting dataset offers hourly temporal resolution and continuous vertical coverage from 0.15 to 6.00 km across 60 altitude layers. Gaussian smoothing was applied to the vertical profiles to suppress high-frequency noise and reduce the impact of outliers, ensuring smoother transitions between adjacent layers and enhancing the physical interpretability of the vertical structure.

### 2.1.2 Observation data

Observations of $PM_{2.5}$ chemical components, including $NH_4^+$, $NO_3^-$, $SO_4^{2-}$, OM, and BC, were measured in situ with an Aerodyne Aerosol Chemical Speciation Monitor (ACSM) in 2021 at an urban site in Beijing. A detailed description of the sampling site and measurements is provided in Sun et al. (2013a). Under well-mixed boundary-layer conditions, $PM_1$ concentrations at 150 m closely match surface values (Sun et al., 2015a; Zhao et al., 2017). Therefore, we directly compared our lowest retrieval level (150 m) with ACSM ground-site measurements throughout 2021, yielding $R^2$ values of 0.70, 0.69, 0.62, 0.61, and 0.58 and root mean square errors (RMSE) of 0.96–7.67 µg/m$^3$ for $NH_4^+$, $NO_3^-$, $SO_4^{2-}$, OM, and BC, respectively, with normalized mean biases (NMB) within ± 0.012 (Fig. S1).

Aircraft observations collected by Liu et al. aboard a KingAir 350 platform during vertical measurement flights (100 m–2.9 km above ground level) were used as input for our chemical retrieval algorithm (Liu et al., 2018). Ambient air was sampled isokinetically and maintained at 650 hPa via a pressure-controlled manifold before being analyzed by a Compact Time-of-Flight Aerosol Mass Spectrometer (C-ToF-AMS, Aerodyne). The AMS provided 1 min–averaged mass concentrations of non-refractory $PM_1$ species, including $NO_3^-$, $SO_4^{2-}$, $NH_4^+$, chloride (Cl$^-$), and organics. In addition, refractory black carbon (rBC) mass concentrations were measured at 1 Hz using a Single Particle Soot Photometer (SP2, Droplet Measurement Technologies) (Liu et al., 2020a). Because aircraft-based measurement campaigns involve substantial organizational, operational, and maintenance costs and are limited to discrete time intervals, continuous and perfectly time-aligned chemical observations with our ground-based lidar retrievals were not available. We therefore selected flight segments under meteorological conditions most closely matching our lidar retrievals to serve as vertical validation data. Comparison of aircraft observations with our retrieved vertical profiles yielded correlation coefficients above 0.92 and RMSEs below 7.9 µg /m$^3$ for all five chemical components (Song et al., 2025), confirming the robustness of our retrieval methodology.

### 2.2 PBLH calculation

Flamant et al. (1997) recognized the planetary boundary layer height (PBLH) as the point where the gradient of Range-Squared Corrected-Signal (RSCS) reaches its minimum, a technique referred to as the first gradient method (GM). This approach is extensively utilized to calculate PBLH using the following formula:

$$h_{GM} = h\left[\min\left(\frac{\partial(RSCS)}{\partial R}\right)\right],\tag{1}$$

For this study, we selected the 1064 nm lidar range-corrected signal strength (RSCS) to retrieve planetary boundary layer height (PBLH) for three reasons. First, the intensity of Rayleigh scattering is inversely proportional to the fourth power of the wavelength, so at 1064 nm the molecular contribution is reduced by a factor of 16 compared to 532 nm, effectively minimizes the contribution from molecular scattering. Second, our Mie lidar's 1064 nm channel is optimally matched to the modal diameter of ambient fine particles, which is approximately 1 μm, maximizing sensitivity to changes in particle loading at the boundary layer top. Third, in our previous study we have successfully applied 1064 nm RSCS for PBLH detection (Wang et al., 2021), demonstrating both its accuracy and reliability. To minimize incomplete overlap effects, the gradient method was applied to the lidar signal at heights above 150 m.

## 2.3 HYSPLIT data

To verify the sources of pollutant transport, we utilized the Hybrid Single-Particle Lagrangian Integrated Trajectory (HYSPLIT) backward trajectory analysis model. This model conducts a systematic analysis of $PM_{2.5}$ source pathways at various stages and altitudes. It identifies the origins and transport routes of air masses through clustering analysis, using resources available at https://www.ready.noaa.gov/HYSPLIT.php.

## 2.4 Other dataset

Nitrogen dioxide ($NO_2$) is a key precursor to particulate nitrate in the atmosphere, and its concentration distribution significantly impacts atmospheric oxidizing capacity and nitrate formation. This study obtained ground-level measurements of $NO_2$ mass concentration at the Beijing Olympic Sports Centre, the nearest site to IAP, available at https://quotsoft.net/air/. The vertical concentration data of $NO_2$ and $O_3$ are sourced from the Copernicus Atmosphere Monitoring Service (CAMS; grid resolution 0.75°×0.75°, temporal resolution 3 h; https://ads.atmosphere.copernicus.eu), which were also used to calculate the atmospheric oxidizing capacity (AOC) as $NO_2 + O_3$. The agreement between $NO_2$ concentrations measured at the Beijing Olympic Sports Center and CAMS ground-level data yields a correlation coefficient of 0.68 (Fig. S2). This less-than-perfect correlation is likely due to strong summer convective mixing and the altitude difference between the two datasets (Cheng et al., 2022; Kuhn et al., 2024).

Meteorological variables, including temperature (T), relative humidity (RH), and wind components (UVW), were obtained from ERA5 hourly data on pressure levels in IAP for 2021, sourced from the European Centre for Medium-Range Weather Forecasts (ECMWF) at https://cds.climate.copernicus.eu/datasets/. The turbulent kinetic energy (TKE) is calculated using the following equation:

$$\text{TKE} = 0.5 \times (\delta_u^2 + \delta_v^2 + \delta_w^2),\tag{2}$$

$$\delta_w^2 = \frac{1}{N-1}\sum_{i=1}^{N}(w_i - \overline{w})^2,\tag{3}$$

$$\delta_u^2 = \frac{1}{N-1} \Sigma_{i=1}^{N} (u_i - \bar{u})^2 \,, \tag{4}$$

$$\delta_v^2 = \frac{1}{N-1} \Sigma_{i=1}^{N} (v_i - \bar{v})^2 \,, \tag{5}$$

where N is the total number of data points obtained from averaging the hourly data over a year every six hours., $w_i$ is the $i^{th}$ vertical wind velocity (m s$^{-1}$), $u_i$ ($v_i$) is the $i^{th}$ horizontal wind speed (m s$^{-1}$), $\bar{w}$ is the mean vertical wind speed (m s$^{-1}$), and $\bar{u}(\bar{v})$ is the mean horizontal wind speed (m s$^{-1}$) (Zhao et al., 2020). The nitrogen oxide ratio (NOR) was calculated by:

$$NOR = \frac{[NO_3^-]}{[NO_3^-] + [NO_2]} \,, \tag{6}$$

where [ ] indicates the mass concentration.

## 3 Results and discussion

### 3.1 General overview

#### 3.1.1 Vertical characteristics

The vertical distribution of nitrate closely mirrored the trend of PM$_{2.5}$, constituting a significant proportion at various altitudes. Throughout 2021, nitrate peaks were observed at 290 m, 900 m, and 1500 m (Fig. 1a), similar to PM$_{2.5}$. Seasonally, across the 0.15-3.00 km, nitrate concentrations in spring, autumn, and winter exceeded those in summer due to the photolytic and volatile nature of nitrates during the summer months (Ye et al., 2017). However, summer still exhibited the aforementioned peaks, which were more pronounced in autumn and winter (Fig. 1b-e). Pollution levels were categorized into six classes: Clean (PM$_{2.5}$ < 35 µg/m$^3$), Light Pollution (35-75 µg/m$^3$), Moderate Pollution (75-115 µg/m$^3$), Heavy Pollution (115-150 µg/m$^3$), Severe Pollution (150-250 µg/m$^3$), and Toxic Pollution (>250 µg/m$^3$). The vertical profiles of particulate nitrate for each pollution category are shown in Figure S3. Within the 3 km altitude range, as pollution levels increased, the nitrate peaks at 290 m and 1500 m gradually diminished, while the peak at 900 m became more pronounced from light to severe pollution levels. This phenomenon was attributed to the boundary-layer structure, in situ formation, long-range transport, etc. Thus, studying the formation and distribution of nitrates at various altitudes is crucial for understanding the vertical accumulation of pollutants.

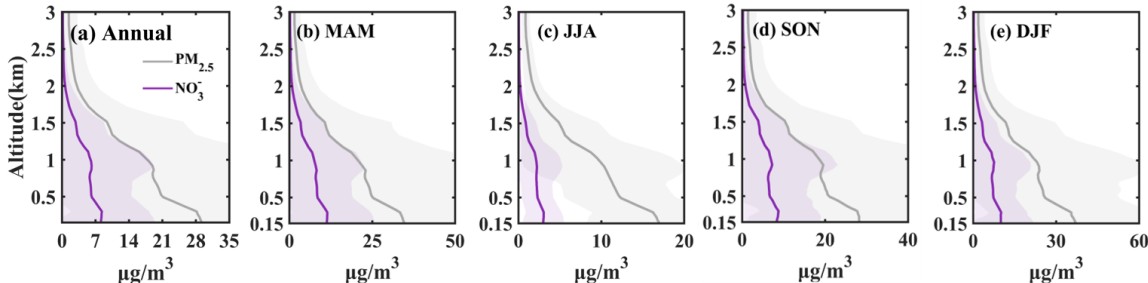

**Figure 1. Profiles of nitrate and PM₂.₅ mass concentrations in 2021. (a) Annual average; (b)-(e) Seasonal averages, with MAM, JJA, SON, and DJF representing spring, summer, autumn, and winter, respectively.**

To elucidate the evolution characteristics of nitrate pollutants in 2021, we conducted a time series analysis of nitrate and PM₂.₅ mass concentrations within a 2 km altitude range, as pollutants were primarily concentrated within the boundary layer. To visually represent the annual trend of nitrates, we applied Gaussian smoothing in the time dimension (Fig. 2). Nitrate concentrations showed two peaks within the 2 km altitude range: from late February to mid-March and from late October to late November. These increases were attributed to frequent agricultural activities that released significant amounts of ammonia

into the atmosphere, which facilitated nitrate formation (Kong et al., 2020). Based on the varying concentration changes at different altitudes, we categorized the analysis into four altitude layers: 0.15-0.30 km, 0.30-0.80 km, 0.80-1.20 km, and 1.20-2.00 km, to comprehensively study the seasonal variations and formation mechanisms of nitrates.

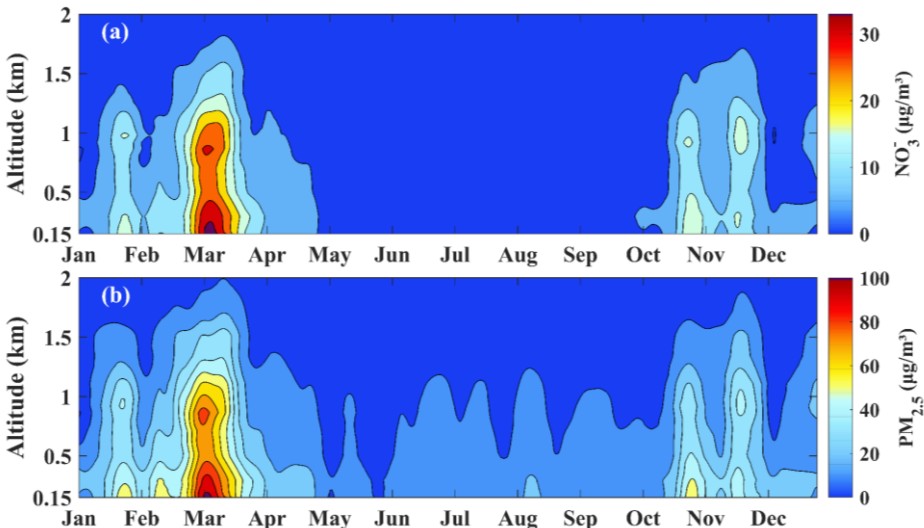

**Figure 2. Vertical profile of nitrate and PM₂.₅ mass concentrations in 2021.**

**3.1.2 Seasonal variation**

Despite nitrates being the dominant component among all PM₂.₅ chemical constituents, their concentrations were lowest in summer across the four altitude layers (Fig. 3). This was attributed to high temperatures and humidity in summer, which facilitated the liquid-phase oxidation of sulfur dioxide but also made nitrates more susceptible to photolysis and volatilization (Gen et al., 2022). Compared to autumn, nitrate levels were slightly higher in winter caused by heating. The ratio of the sum

of sulfates, nitrates, and ammonium to PM₂.₅ (SNA/PM₂.₅) increased with altitude (Fig. 3e-g), consistent with previous studies conducted at the Canton Tower in Guangzhou (Zhou et al., 2020). The ratio of nitrates to sulfates ($NO_3^-/SO_4^{2-}$) indicated emission sources, distinguishing between mobile sources (e.g., traffic, urbanization, biomass burning) and stationary sources (e.g., industrial activities, fossil fuel power generation) (Li et al., 2020). $NO_3^-/SO_4^{2-}$ was higher in spring and autumn due to

the preferential formation of sulfates and the saturation of sulfates during these seasons (Cao et al., 2012), while it was lower in summer and winter (Fig. 3a-d), reflecting seasonal variations in pollution source intensity. Spring and autumn were peak seasons for agricultural activities (such as fertilization and straw burning), contributing nitrogen sources (e.g., ammonia) that promoted further nitrate formation (Sun et al., 2022). Furthermore, the high pollution levels observed in summer and winter were mainly attributed to stationary sources, as indicated by the low value of $NO_3^-/SO_4^{2-}$. Therefore, variations in nitrate concentrations reflected the interplay of human activities, meteorological factors, and chemical processes.

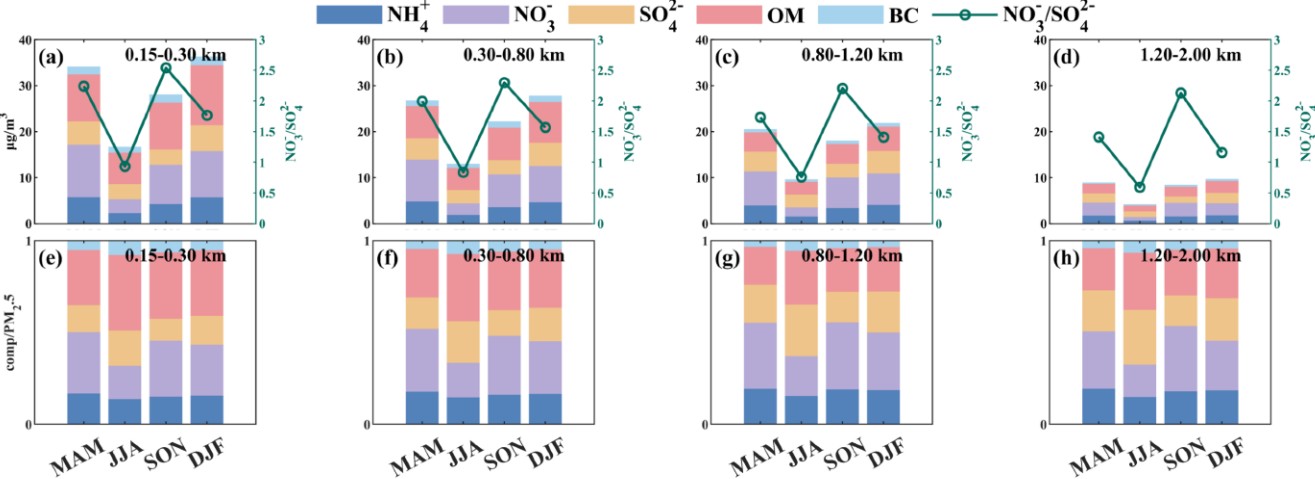

Figure 3. Seasonal variations of mass concentrations (a-d) and mass fractions (e-h) of black carbon (BC), organic matter (OM), sulfate ($SO_4^{2-}$), nitrate ($NO_3^-$), and ammonium ($NH_4^+$). The green line represents the ratio of nitrate to sulfate ($NO_3^-/SO_4^{2-}$).

The daily variation in nitrate concentrations at different altitude levels was significant throughout the seasons (Fig. 4). In spring, daytime nitrate levels exceeded nighttime concentrations, with this trend diminished with altitude. This pattern correlated with increased $NO_2$ emissions during peak traffic hours, evidenced by a high correlation coefficient (0.79-0.85) for nitrates and $NO_2$ below 1.20 km. Intense photochemical reactions in summer primarily converted $NO_2$ into $O_3$ and hydroxyl radicals ($HO_x$), which ultimately led to the formation of nitric acid ($HNO_3$), particulate nitrate, and various organic nitrates. During summer, high temperatures and low relative humidities shifted the $NH_4NO_3$ equilibrium toward the gas phase, and a deeper convective boundary layer further diluted surface aerosols. Consequently, particulate nitrate concentrations were lower than in other seasons and exhibited only a weak correlation with $NO_2$ (R values ranged from 0.13 to 0.48). Daytime nitrate levels in summer surpassed those at night due to photochemical processes. In autumn, daily nitrate variation was less pronounced, yet it exhibited a strong correlation with $NO_2$ (R values ranged from 0.80 to 0.92). Notably, a multi-peak structure appeared at 0.80-1.20 km, with the highest R value of 0.92 observed in this range. Nonetheless, in terms of daily variation, the average nitrate concentrations in autumn were significantly lower than those in spring. Winter showed a distinct daily pattern characterized by high nitrate concentrations in the afternoon and nighttime, with lower levels before noon. This trend was most evident at 0.80-1.20 km and disappeared above 1.20 km. Three notable peaks in nitrate concentrations occurred in winter at 13:00, 18:00,

and 22:00, which were attributed to the accumulation of photochemically generated nitrates in the residual layer and the increasing oxidation during the winter night, consistent with the conclusions drawn by previous researchers (Yang et al., 2021b; Zang et al., 2022).

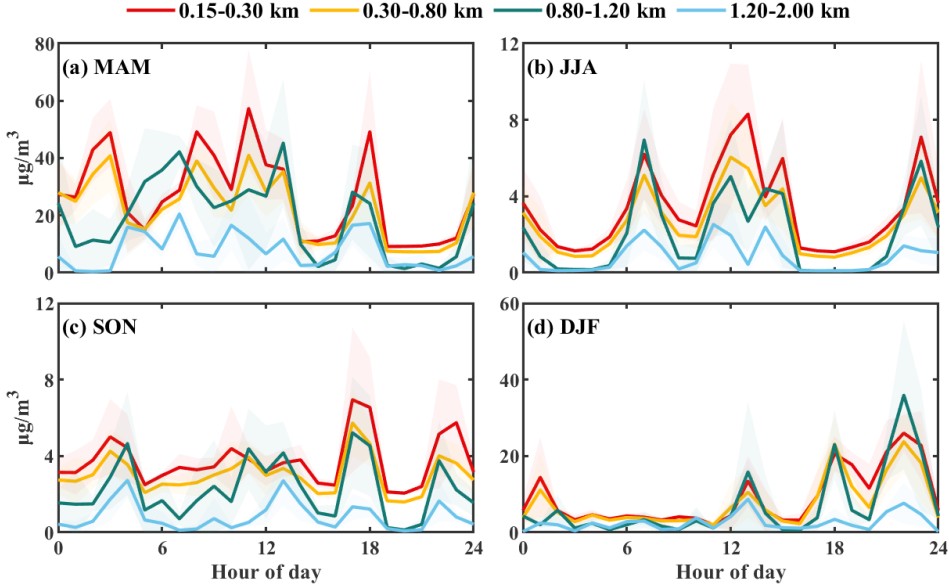

**Figure 4. Diurnal variation of nitrate mass concentrations at four height levels during the four seasons in 2021.**

### 3.2 Seasonal formation mechanisms

To better understand the seasonal variations in nitrate formation, this study categorized the mechanisms into three types: thermodynamics, photochemistry, and dynamics, aligning with the classification method of Ge et al. (2017). The thermodynamic mechanism was characterized by RH and T to assess the contribution of heterogeneous reactions to nitrate formation. The photochemical mechanism was represented by $O_3$ concentration and AOC to evaluate the impact of photochemical reactions. The dynamic mechanism was characterized by TKE and vertical wind speed (w) to analyze the influence of dynamics on nitrate formation. We conducted correlation analyses between nitrate concentrations and these factors (Fig. S4), with results shown in Figure 5. We categorized the correlation coefficients between nitrate mass concentrations and various factors by season and altitude, revealing distinct nitrate formation mechanisms across different conditions. In the 0.15-0.30 km range, RH showed a significant positive correlation with nitrate concentrations in spring, summer, and winter (R values ranged from 0.43 to 0.68, $p < 0.01$), while temperature exhibited a significant negative correlation in spring (R = –0.76, $p < 0.01$) and summer (R = –0.10, $p < 0.01$). These results indicated that thermodynamic factors were the primary drivers of increased nitrate concentrations in Beijing's lower atmospheric layers during these seasons.

For photochemical factors, both AOC and $O_3$ exhibited positive correlations with nitrate concentrations in winter and summer, consistent with Zhang et al. (2024). However, they exhibited a negative correlation in spring and autumn. The positive

correlation between temperature and nitrate in summer and winter further supported the contribution of photochemical processes. In winter, the coexistence of thermodynamic and photochemical factors explained the peaks in nitrate concentrations at three specific times during the winter daily variation mentioned in section 3.1.1. Regarding dynamic factors, TKE was negatively correlated with nitrate in spring and autumn, suggesting it hindered nitrate diffusion. In spring, nitrate negatively correlated with w, while in autumn, a positive correlation appeared at altitudes of 1.20-2.00 km. This suggested that dynamic factors contributed differently in spring and autumn. Vertical wind speed in autumn facilitated nitrate vertical transport, whereas in spring, it aided diffusion. Notably, in spring and autumn, the negative correlation between RH and nitrate concentrations at higher altitudes suggested that increased aerosol liquid water under humid conditions enhanced the growth of particle nitrate. Those larger, more hygroscopic particles settled more rapidly, leading to lower nitrate levels in the upper layers of the atmosphere.

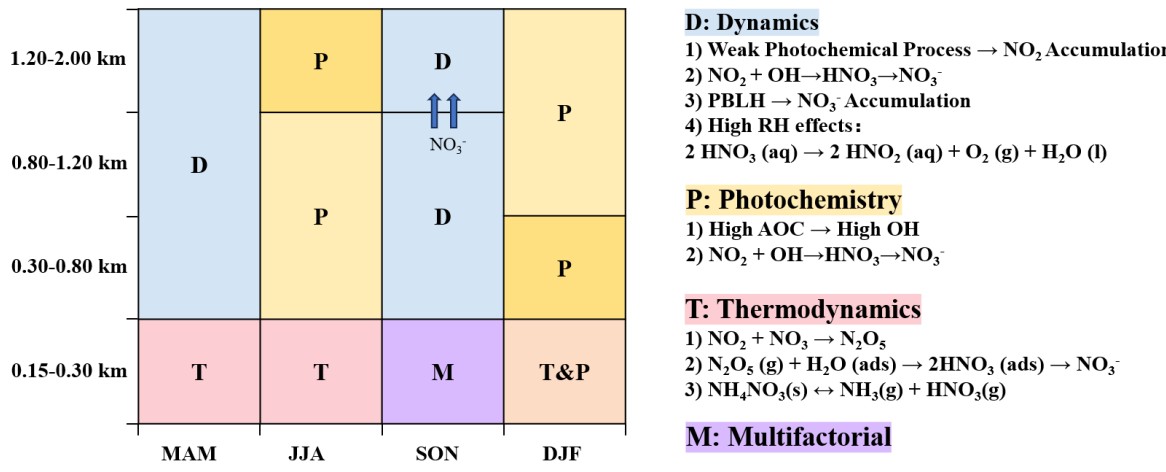

**Figure 5. Vertical distribution of nitrate formation drivers across seasons.**

## 3.3 Case studies

Based on the overall analysis of seasonal variations in annual nitrate concentration, we classified the seasonal formation mechanisms of nitrate at different heights. This section will provide a detailed analysis of each formation mechanism in relation to specific pollution processes.

### 3.3.1 Thermodynamics-driven case

From $25^{th}$ to $28^{th}$ March, the hourly concentrations of ground-level $PM_{2.5}$ and nitrate reached 177.66 μg/m$^3$ and 72.59 μg/m$^3$, respectively. Nitrate was predominantly concentrated within the urban boundary layer, with the highest pollution levels observed at the surface and the top of the boundary layer (Fig. 6i-j). At 0.15-0.30 km, nitrate showed a positive correlation with RH and a negative correlation with T (Fig. 6a, 6e). This indicated that this layer was a typical thermally driven layer,

where low temperatures and high humidity created favorable conditions for the heterogeneous reaction of $N_2O_5$ to form nitrate

(Wang et al., 2018). As altitude increased, the influence of thermal driving became less significant.

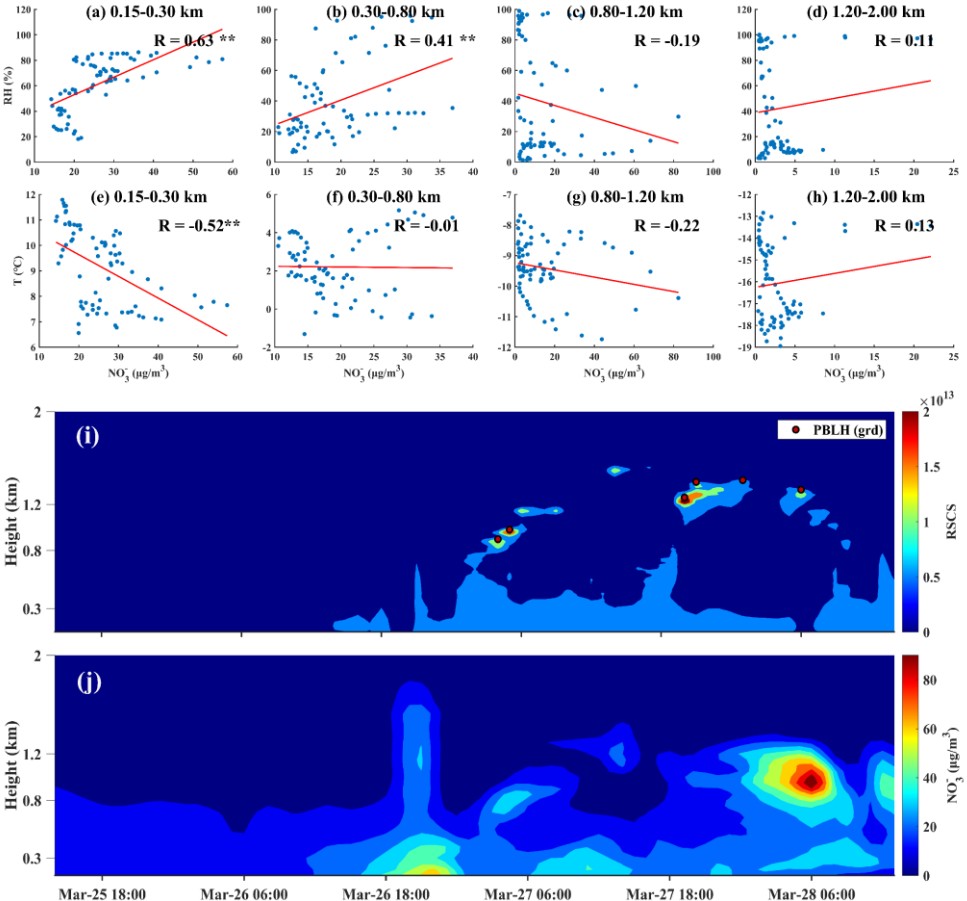

Figure 6: (a-d): Correlation between nitrate mass concentration and RH at various altitude levels (** indicates p < 0.01); (e-h): Correlation between nitrate mass concentration and T at various altitude levels; (i): Lidar range-squared corrected signal and planetary boundary layer height derived using the gradient method; (j): Nitrate mass concentration profile.

**3.3.2 Dynamics-driven case**

From 28$^{th}$ February to 4$^{th}$ March, A significant event of high nitrate concentration was detected at the top of the boundary layer in Beijing, specifically within the altitude range of 0.6 to 1.3 km (Fig. 7c-d). On 1$^{st}$ March, in the afternoon, PM$_{2.5}$ and nitrate levels peaked at 273.03 µg/m³ and 87.43 µg/m³, respectively, indicating a severe upper-air pollution event. The correlation

coefficient between nitrate and NOR was 0.86, indicating that the increase in nitrogen oxides likely promoted the formation of nitrate. However, the correlation between nitrate and AOC was -0.02, suggesting that atmospheric oxidation had minimal contribution to this pollution event. Furthermore, a correlation of 0.54 with turbulent kinetic energy (TKE) suggested that increased turbulence played a crucial role in enhancing the mixing and distribution of particulates, resulting in a more uniform

atmospheric concentration of nitrates. Turbulence also accelerated the reactions of gaseous precursors, increasing nitrate production rates. Backward trajectory analysis conducted at 14:00 (LST) on 1st March revealed that polluted air masses from the northwest were transported to Beijing (Fig. 7a). Prevailing westerly winds facilitated this transport. During the pollution event, upward airflow below 0.8 km lifted pollutants, allowing them to accumulate in the upper atmosphere. The strong upward flow above 0.8 km on 2nd March at 00:00 led to the dissipation of nitrate pollution (Fig. 7b). Thus, the accumulation of pollutants and precursors from the northwest, combined with dynamic atmospheric conditions, resulted in this severe upper-air pollution event.

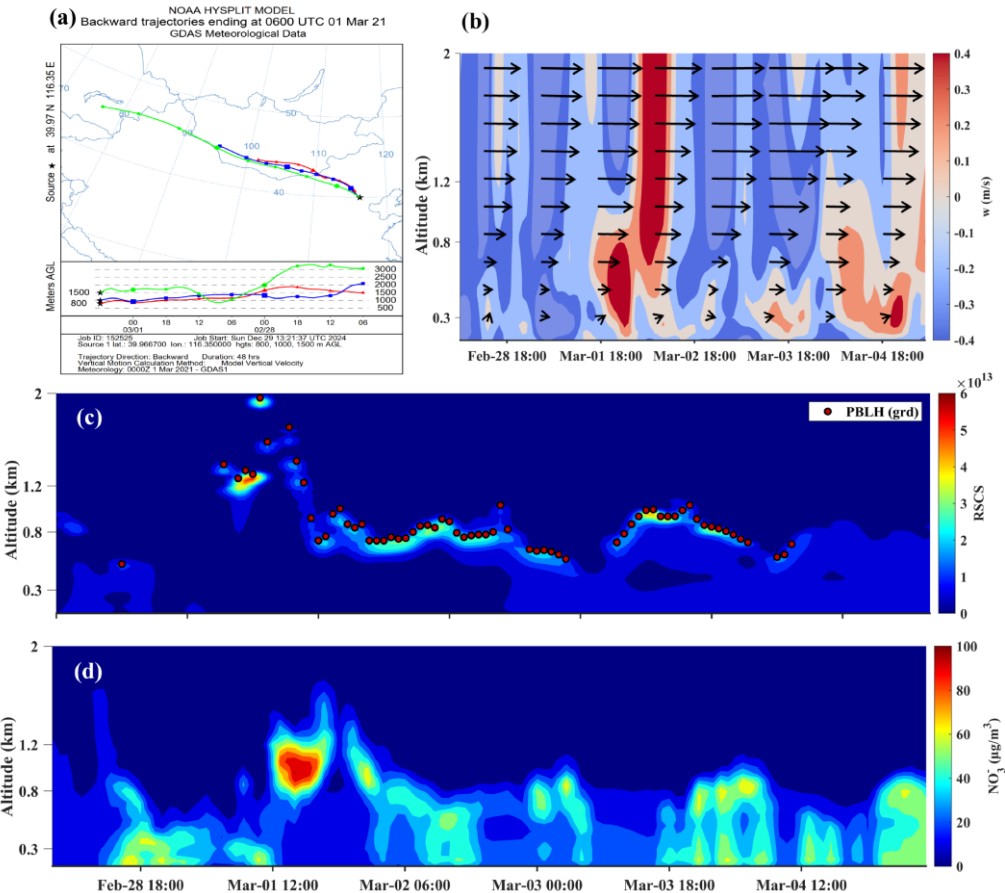

**Figure 7. (a) Back trajectory map for 21st March at 14:00 (LST). (b) Vertical wind speed profile and horizontal wind vector diagram. (c) Lidar range-squared corrected signal and planetary boundary layer height from 28th February to 5th March, derived using the gradient method. (d) Nitrate mass concentration profile.**

### 3.3.3 Photochemistry-driven case

From 28th to 29th November, significant high-altitude nitrate pollution was also recorded within the urban boundary layer. This event was closely linked to meteorological conditions, chemical reactions, and human activities. Notably, PM$_{2.5}$ concentrations surged to 278.70 μg/m³ at an altitude of 0.8-1.2 km, while nitrate levels reached an alarming 118.11 μg/m³. Compared to

previous pollution events, this incident exhibited both a higher severity and a longer duration (Fig. 8). Within altitudes below
2 km, a strong positive correlation was observed between nitrate concentration and AOC (Fig. 8a-d) while a negative
correlation with $O_3$ was noted below 1.2 km (Fig. 8e-h). In addition, we compared the vertical profiles of AOC, $NO_2$, and $O_3$
(Fig. S5) and found that the AOC and $NO_2$ profiles closely mirrored the nitrate vertical structure shown in Figure 8d, whereas
the $O_3$ profile did not. This indicated that under conditions of high AOC, $NO_2$ was more likely to be converted to nitrates rather
than $O_3$. This conversion was influenced by various environmental factors, such as temperature, humidity, and light conditions,
with AOC playing a crucial role in this high-altitude pollution event. Furthermore, the impact of AOC on pollution events in
winter should not be underestimated, as it has become a primary driver of high-altitude nitrate pollution (Fig. 9). Therefore, to
control and reduce high-altitude nitrate pollution effectively, a comprehensive understanding and management of atmospheric
oxidative capacity is essential for developing effective pollution control strategies.

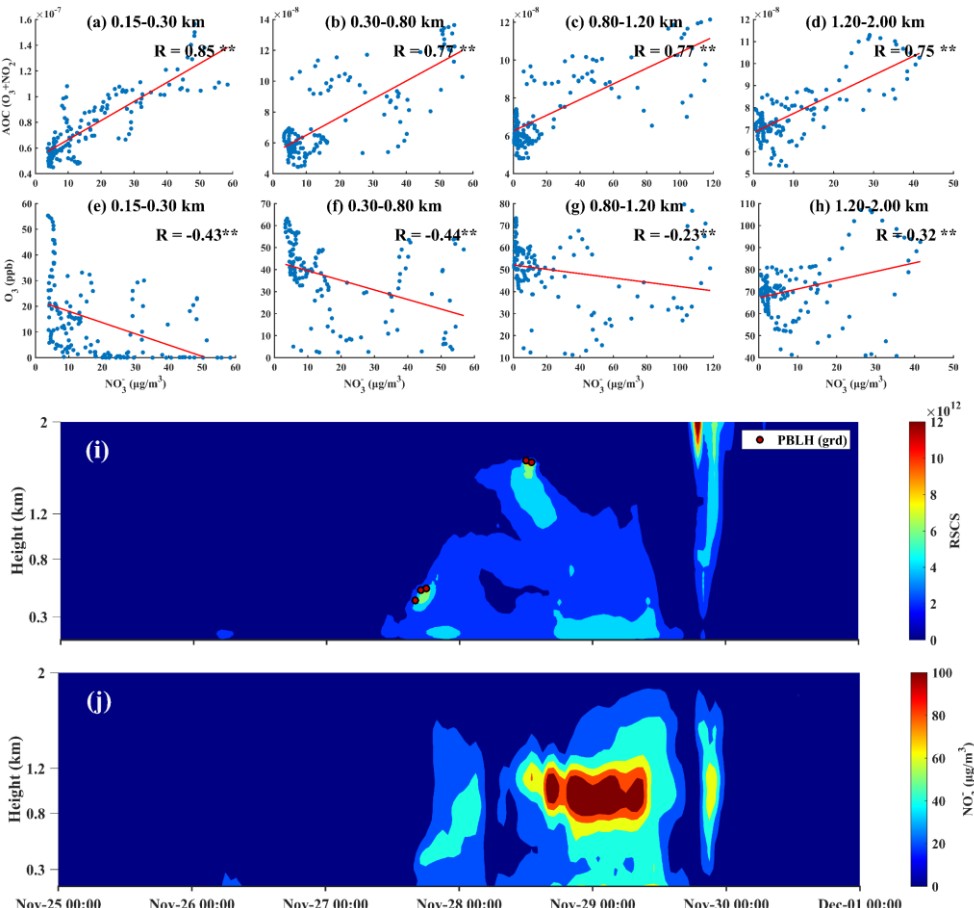

**Figure 8. (a-d): Correlation between nitrate mass concentration and AOC at various altitude levels (** indicates $p < 0.01$); (e-h): Correlation between nitrate mass concentration and $O_3$ at various altitude levels; (i): Lidar range-squared corrected signal and planetary boundary layer height derived using the gradient method; (j): Nitrate mass concentration profile.**

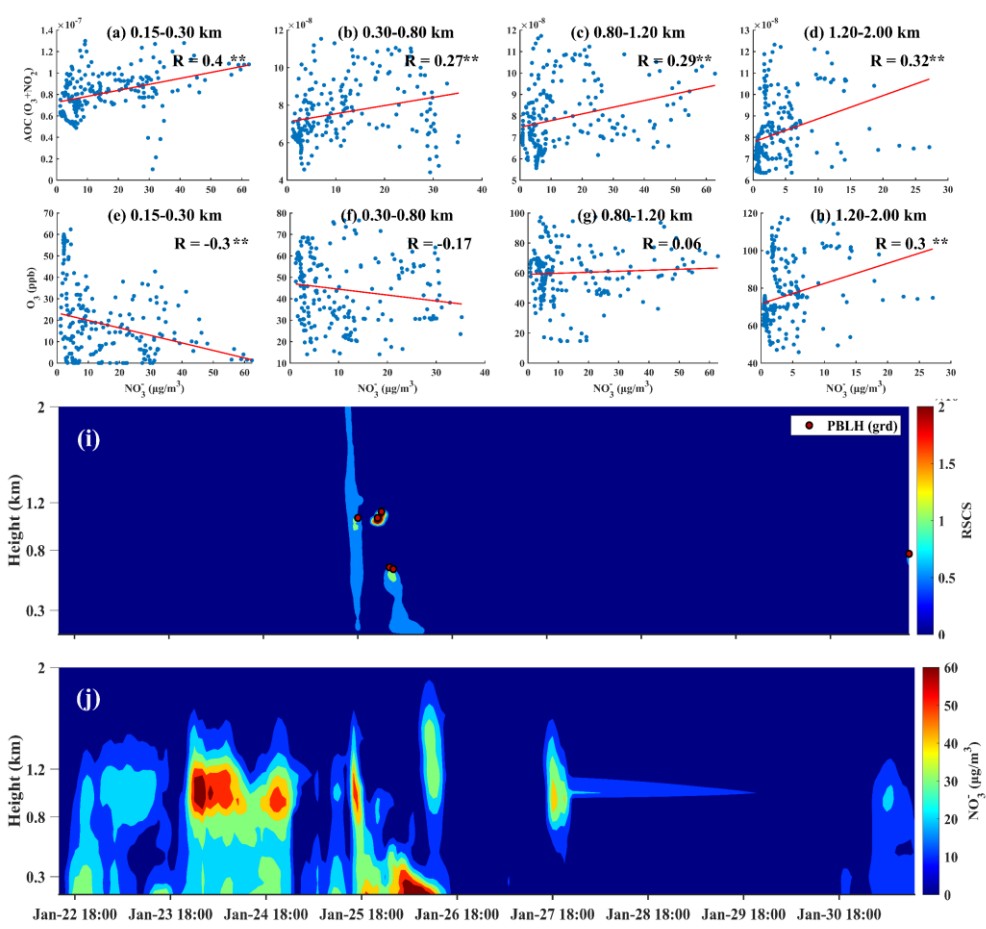

Figure 9. Same as Figure 8, but for 22$^{nd}$ -30$^{th}$ January.

## 4 Conclusions

This study comprehensively examined the vertical distribution, seasonal variation, and formation mechanisms of nitrate pollutants in urban Beijing. Vertical profiling revealed a close correlation between nitrate levels and PM$_{2.5}$ concentrations, suggesting common characteristics in the decline of air quality. Key findings showed significant fluctuations in nitrate mass concentrations at various altitudes, with elevated levels observed during early spring and late autumn. This was attributed to ammonia emissions from agricultural activities during these periods, as ammonia served as a precursor for nitrate formation. Additionally, early spring's high aerosol pH, influenced by dust events, enhanced the conversion of NO$_x$ to nitric acid (HNO$_3$). High concentrations of nitrate pollution in the upper atmosphere during early spring were primarily due to warming temperatures and increased atmospheric oxidizing capacity. However, over the entire spring season, dynamic processes predominantly drove nitrate formation. In late autumn, Beijing experienced heightened industrial and transportation activities, coinciding with concentrated straw burning. The NO$_x$ emissions from straw burning contributed to nitrate pollution. Unlike

spring, high nitrate concentrations in the upper atmosphere during late autumn were not attributed to photochemical reactions. Instead, they resulted from the transport of polluted air masses from the northwest, which were then lifted to higher altitudes by upward air currents.

Winter exhibited the highest nitrate concentrations, with notable diurnal variations that peaked at 13:00, 18:00, and 22:00. Peak concentrations rose with altitude up to 1.20 km, primarily due to photochemical reactions. At night, atmospheric oxidants and pollutants from these reactions accumulated in the residual layer, enhancing nighttime atmospheric oxidation and increasing nitrate production, especially between 0.3 and 0.8 km. In summer, increased temperature and humidity enhanced the photolysis and volatilization of nitrates, resulting in the lowest overall concentrations. However, the formation of nitrates

in the mid and upper layers continued to be dominated by photochemical processes, with these reactions being most pronounced at 1.20-2.00 km. Overall, nitrate levels in both winter and mid-summer showed a positive correlation with atmospheric oxidative capacity, highlighting the significance of photochemical processes during these seasons. In contrast, thermodynamic mechanisms were found to have a more substantial impact on nitrate formation at lower altitudes (0.15-0.30 km).

The study analyzed the differing roles of thermodynamic, dynamic, and photochemical mechanisms in causing nitrate pollution at various altitudes, emphasizing the severity and underlying mechanisms of high-altitude nitrate pollution. These findings underscore the importance of understanding meteorological conditions and human activities in managing nitrate pollution at different heights. Given the critical role of atmospheric oxidative capacity in high-altitude nitrate pollution events, prioritizing its monitoring and management is essential. These insights are vital for designing effective air quality control strategies,

particularly in urban areas with high pollution risks.

**Data availability**

The data in this study are available from the authors upon request (tingyang@mail.iap.ac.cn).

**Supplement**

The supplement related to this article is available online at:

**Author contribution**

YT performed the analysis, visualized the data and wrote the original manuscript. TY provided scientific guidance, designed the paper structure and revised the manuscript. HL provided the components data. PT provided the measurement data. All authors reviewed and revised this paper.

## Competing interests

The authors declare that they have no conflict of interest.

## Disclaimer

Publisher's note: Copernicus Publications remains neutral with regard to jurisdictional claims in published maps and institutional affiliations.

## Acknowledgements

This work was supported by the National Key Research and Development Program for Young Scientists of China (No. 2022YFC3704000), National Natural Science Foundation of China (NSFC) Excellent Young Scientists Fund (No. No. 42422506), the National Natural Science Foundation of China (No. 42275122), We thank for the technical support of the National large Scientific and Technological Infrastructure "Earth System Numerical Simulation Facility" (https://cstr.cn/31134.02.EL). Ting Yang would like to express gratitude towards the Program of the Youth Innovation Promotion Association (CAS).

## Financial support

This work was supported by the National Natural Science Foundation of China (No. 42422506, No. 42275122) and the National Key Research and Development Program for Young Scientists of China (grant no. 2022YFC3704000)

## Review statement

This paper was edited by Suvarna Fadnavis and reviewed by two anonymous referees.

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
