# Peer review of "Vertically Resolved Formation Mechanisms of Fine Particulate Nitrate in Asian Megacities: Integrated Lidar – Aircraft Observations and Process Analysis"

_EGUsphere, 2025_

## Author Comment (AC1)

**Authors' responses to Referees' comments**

**Journal:** Atmospheric Chemistry and Physics (ACP)

**Manuscript Number:** ACP-2025-898

**Title:** Vertically Resolved Formation Mechanisms of Fine Particulate Nitrate in Asian Megacities: Synergistic Lidar-Aircraft Observations and Process-Based Analysis

**Authors:** Yutong Tian, Ting Yang, et al.

Note:

Comment (13-point black italicized font).

Reply (indented, 13-point blue normal font).

"Revised text as it appears in the text (in quotes, 12-point blue italicized font)".
* * *
**Anonymous Referee #2**

*1 General comments:*

*The manuscript addresses an important environmental issue, vertical formation mechanism of nitrates in Asian megacities. It is overall well written, providing important progress through vertically continuous observations and comprehensive data analysis. My specific comments are as follows.*

**Authors' response:**

We thank the reviewer for the positive assessment and constructive suggestions of our manuscript.

*2 Detailed Comments:*

*1) It's recommended to change the title from "Synergistic Lidar - Aircraft Observations and Process - Based Analysis" to "Integrated Lidar - Aircraft Observations and Process Analysis".*

**Authors' response:**

Thanks for your suggestion. Accordingly, we have revised the title from "Synergistic Lidar – Aircraft Observations and Process-Based Analysis" to "Integrated Lidar – Aircraft Observations and Process Analysis" to improve clarity and conciseness.

*2) In the Data and Methods section, include more details about the machine learning*

*model (such as algorithm name, input variables, R2 value for validation accuracy) and cite relevant studies.*

**Authors' response:**

Thanks for your suggestion. In the revised manuscript we have substantially expanded the Data and Methods section to specify that we use a hybrid convolutional neural network (CNN)–long short-term memory (LSTM) mapping model (Hinton et al., 2006; Wang et al., 2016) whose hyperparameters are automatically tuned via Bayesian optimization (Frazier, 2018). We now clearly list all twelve input variables—six optical retrievals (AN, AW, WSOM, WIOM, BC, extinction at 532 nm) plus geopotential height, relative humidity, temperature, and the u, v, w wind components—and state that the model outputs mass concentrations of $NH_4^+$, $NO_3^-$, $SO_4^{2-}$, OM and BC with hourly resolution and 60 vertical layers (0.15–6.00 km).

Validation against independent ground-truth measurements from an Aerodyne Aerosol Chemical Speciation Monitor (ACSM) yields $R^2$ values for $NH_4^+$, $NO_3^-$, $SO_4^{2-}$, OM and BC of 0.70, 0.69, 0.62, 0.61, and 0.58, respectively, with root-mean-square errors of 0.96-7.67 µg m$^{-3}$ and normalized mean biases within 0.012(Figure R1). These results are provided in the Supplementary Information. Moreover, in our previous publications, we have validated these vertical profiles against aircraft measurements (Song et al., 2025; Tan et al., 2025), achieving correlation coefficients above 0.94 and root mean square errors below 7.9, and demonstrating a strong agreement between the retrieved profiles and airborne observations as Figure R2.

[Figure]

**Figure R1. Validation of ground-layer retrieval results against ACSM observations.**

[Figure]

**Figure R2. Vertical profile validation against 2016 aircraft measurements: 2022 Winter Olympics (a); 2021–2022 (b).**

**Reference**

Frazier, P.: A Tutorial on Bayesian Optimization, ArXiv, abs/1807.02811, 2018.

Hinton, G. E., Osindero, S., and Teh, Y.-W.: A Fast Learning Algorithm for Deep Belief Nets, Neural Computation, 18, 1527-1554, 10.1162/neco.2006.18.7.1527, 2006.

Song, Y., Yang, T., Tian, P., Li, H., Tian, Y., Tan, Y., Sun, Y., and Wang, Z.: Novel Insights into the Vertical Distribution Patterns of Multiple PM2.5 Components in a Super Mega-City: Responses to Pollution Control Strategies, 10.3390/rs17071151, 2025.

Tan, Y., Yang, T., Li, H., Tian, P., Song, Y., He, J., Tian, Y., Sun, Y., and Wang, Z.: Unveiling the vertical dynamics of atmospheric ammonium in Asia megacity: A GRASP-based investigation of spatiotemporal patterns and source drivers, Atmospheric Research, 325, 108201, https://doi.org/10.1016/j.atmosres.2025.108201, 2025.

Wang, M., Song, L., Yang, X., and Luo, C.: A parallel-fusion RNN-LSTM architecture for image caption generation, 2016 IEEE International Conference on Image Processing (ICIP), 25-28 Sept. 2016, 4448-4452,    10.1109/ICIP.2016.7533201,

*3) Line 92,* plea*se add a bit more information about Gaussian smoothing.*

**Authors' response:**

Thanks for your suggestion. In response, we have expanded the description in Line 92 to read: "*Gaussian smoothing was applied to the vertical profiles to suppress high-frequency noise and reduce the impact of outliers, ensuring smoother transitions between adjacent layers and enhancing the physical interpretability of the vertical structure.*" This added detail clarifies how the smoothing improves the robustness and readability of our retrievals.

*4) Line103, explain why 1064 nm wavelength was selected.*

**Authors' response:**

Thanks for your suggestion. In response, we have expanded the text at Line 103 to read: "*For this study, we selected the 1064 nm lidar range-corrected signal strength (RSCS) to retrieve planetary boundary layer height (PBLH) for three reasons. First, the intensity of Rayleigh scattering is inversely proportional to the fourth power of the wavelength, so at 1064 nm the molecular contribution is reduced by a factor of 16 compared to 532 nm, effectively minimizes the contribution from molecular scattering. Second, our Mie-lidar's 1064 nm channel is optimally matched to the ~1 μm modal diameter of ambient fine particles, maximizing sensitivity to changes in particle loading at the boundary layer top. Third, in our previous study we have successfully applied 1064 nm RSCS for PBLH detection (Wang et al., 2021), demonstrating both its accuracy and reliability. To avoid incomplete-overlap artifacts, we then applied the gradient method to the RSCS profile above 150 m.*" This addition clarifies our rationale for choosing 1064 nm and highlights its suitability for PBLH detection.

**Reference**

Wang, F., Yang, T., Wang, Z., Chen, X., Wang, H., and Guo, J.: A comprehensive evaluation of planetary boundary layer height retrieval techniques using lidar data under different pollution scenarios, Atmospheric Research, 253, 105483, https://doi.org/10.1016/j.atmosres.2021.105483, 2021.

*5) Line 117, please add the grid resolution of CAMS data.*

**Authors' response:**

Thanks for your suggestion. We have updated Line 117 to read: "*The vertical concentration data of $NO_2$ and ozone ($O_3$) are sourced from the Copernicus Atmosphere Monitoring Service (CAMS; grid resolution 0.75° × 0.75°, temporal resolution 3 h; https://ads.atmosphere.copernicus.eu), which were also used to calculate the atmospheric oxidizing capacity (AOC) as $NO_2$+ $O_3$.*" This addition clarifies the spatial and temporal resolution of the CAMS data used.

*6) Line 140, The "Toxic Pollution" category (>250 μg/m³) mentioned in the text seems*

*to have no data support in Fig. 1. Perhaps either delete it or merge it into other categories.*

**Authors' response:**

Thanks for your suggestion. We have added the "Toxic Pollution" category (>250 μg/m³) and relocated all pollution-level–based vertical profile panels to the Supplementary Materials as Figure R3, thereby ensuring that the main figures remain concise while fully supporting each category mentioned in the text.

[Figure]

**Figure R3. Vertical profiles of PM2.5 and nitrate mass concentrations under different pollution conditions.**

*7) Line 140-143, Peak height correlates with the boundary layer structure. Does the peak at 900-m reach the top of the inversion layer?*

**Authors' response:**

Thanks for your suggestion. The atmospheric boundary-layer top was calculated using the gradient method, so its height closely approximates the inversion layer top. Comparing the 900 m concentration peak height with the mean planetary boundary-layer height (PBLH) under each pollution scenario (Table R1) shows that PBLH exceeds 900 m except during serious pollution, when it falls below that level. Under clean to heavy polluted conditions, 900 m lies within the boundary layer and the

observed concentration maximum results from the inversion inhibiting vertical dispersion. However, during severe pollution, 900 m is above the boundary-layer top (inversion), indicating that the $PM_{2.5}$ or nitrate peak arises from the combined effects of secondary in situ formation, vertical transport within the boundary layer, and long-range horizontal advection aloft, producing a localized maximum.

Table R1. Mean PBLH under different pollution conditions

| Pollution Level | Clean | Light | Moderate | Heavy | Severe |
|---|---|---|---|---|---|
| Mean PBLH (m) | 1381.4 | 1101.1 | 1058.2 | 965.7 | 700.2 |

*8) Figure 5: too much information. Please split and simplify the figure.*

**Authors' response:**

Thank you for this suggestion. We have divided the original Figure 5 into two parts: the correlation panel (formerly Figure 5a) has been redrawn as Figure R4 and relocated to the Supplementary Materials, and the conceptual panel has been streamlined as Figure R5 by removing redundant annotations and excessive text to enhance clarity.

[Figure]

**Figure R4. Heatmap of correlations between nitrate and each driven factor across four seasons and four height layers, with blank cells for non-significant correlations.**

[Figure]

**Figure R5. Vertical distribution of nitrate formation drivers across seasons.**

*9) Figure 8: Please add vertical profiles of AOC and NO₂/O₃ to support the conclusion that "high AOC promotes NO₂ to nitrate, not O₃.".*

**Authors' response:**

Thank you for this valuable suggestion. We have added vertical profiles of AOC, $NO_2$, and $O_3$ as a supplementary figure (Figure R6) in the Supplementary Materials. Below 2 km, the AOC and $NO_2$ profiles closely mirror the nitrate vertical structure shown in Figure 8, whereas the $O_3$ profile does not, thereby reinforcing our conclusion that high AOC preferentially drives $NO_2$ conversion to nitrate rather than to $O_3$.

[Figure]

**Figure R6. Vertical profiles of NO₂, O₃, and AOC from November 25 to 30, 2021.**

*10) Figure 1,4 A bit too many panels. The authors could move the panels f - k to the supplementary materials.*

**Authors' response:**

Thanks for your suggestion. For Figure 1, we have moved panels f–k as Figure R3 to the Supplementary Materials to enhance clarity. As for Figure 4, we have consolidated the 16 panels into 4 by integrating four height layers for each season into a single panel (Figure R7), which allows for a more concise and clear depiction of the information.

[Figure]

**Figure R7. Diurnal variation of nitrate mass concentrations at four height levels during the four seasons in 2021.**

*11) The number of equations seems odd. Please double check.*

**Authors' response:**

Sorry for the typo. We have corrected it.

*12) Tense check: Please use past tense for the results discussion.*

**Authors' response:**

Thanks for your suggestion. I have adjusted tense of the conclusion in the revised manuscript.

*13) Make sure all citations are listed. For example, Guan et al., 2024 is cited in the introduction but not in the references.*

**Authors' response:**

I'm sorry for the unclear style of reference list. The Guan 2024 have been included in the list. I have changed the style of list to show the reference information clearly.

---

## Author Comment (AC2)

**Authors' responses to Referees' comments**

**Journal:** Atmospheric Chemistry and Physics (ACP)

**Manuscript Number:** ACP-2025-898

**Title:** Vertically Resolved Formation Mechanisms of Fine Particulate Nitrate in Asian Megacities: Synergistic Lidar-Aircraft Observations and Process-Based Analysis

**Authors:** Yutong Tian, Ting Yang, et al.

Note:

Comment (18-point black italicized font).

Reply (indented, 18-point blue normal font).

"Revised text as it appears in the text (in quotes, 18-point blue italicized font)".
* * *
**Anonymous Referee #3**

*1 General comments:*

*This work comprehensively investigated the vertical distribution, seasonal variation, and formation mechanisms of nitrate pollution at different altitudes in urban Beijing, and thus is important. There are some comments which require to be addressed before it can be accepted.*

**Authors' response:**

We thank the reviewer for the positive assessment and constructive suggestions of our manuscript.

*2 Detailed Comments:*

*1) The title showed aircraft observation, however there is no any description in the main text. Add necessary related description or remove the aircraft from the title.*

**Authors' response:**

Thanks for your suggestion. We have added a clear, concise description of the aircraft observations, including the sampling platform, instrumentation, and validation results (Figure R1) ,to Section 2.1.2 (Observation Data) to align the title with the text:

*"Aircraft observations collected by Liu et al. aboard a KingAir 350 platform during vertical measurement flights (100 m–2.9 km above ground level) were used as*

*input for our chemical retrieval algorithm (Liu et al., 2018). Ambient air was sampled isokinetically and maintained at 650 hPa via a pressure-controlled manifold before being analyzed by a Compact Time-of-Flight Aerosol Mass Spectrometer (C-ToF-AMS, Aerodyne). The AMS provided 1 min–averaged mass concentrations of non-refractory $PM_1$ species, including nitrate ($NO_3^-$), sulfate ($SO_4^{2-}$), ammonium ($NH_4^+$), chloride ($Cl^-$), and organics. In addition, refractory black carbon (rBC) mass concentrations were measured at 1 Hz using a Single Particle Soot Photometer (SP2, Droplet Measurement Technologies) (Liu et al., 2020). Because aircraft-based measurement campaigns involve substantial organizational, operational, and maintenance costs and are limited to discrete time intervals, continuous and perfectly time-aligned chemical observations with our ground-based lidar retrievals were not available. We therefore selected flight segments under meteorological conditions most closely matching our lidar retrievals to serve as vertical validation data. Comparison of aircraft observations with our retrieved vertical profiles yielded correlation coefficients (R) above 0.92 and root-mean-square errors (RMSE) below 7.9 µg m$^{-3}$ for all five chemical components (Song et al., 2025), confirming the robustness of our retrieval methodology."*

[Figure]

**Figure R1. Comparison of aircraft observations with retrieved vertical profiles.**

**Reference**

Liu, Q., Ding, D., Huang, M., Tian, P., Zhao, D., Wang, F., Li, X., Bi, K., Sheng, J.,

Zhou, W., Liu, D., Huang, R., and Zhao, C.: A study of elevated pollution layer over the North China Plain using aircraft measurements, Atmospheric Environment, 190, 188-194, https://doi.org/10.1016/j.atmosenv.2018.07.024, 2018.

Liu, D., Hu, K., Zhao, D., Ding, S., Wu, Y., Zhou, C., Yu, C., Tian, P., Liu, Q., Bi, K., Wu, Y., Hu, B., Ji, D., Kong, S., Ouyang, B., He, H., Huang, M., and Ding, D.: Efficient Vertical Transport of Black Carbon in the Planetary Boundary Layer, Geophysical Research Letters, 47, e2020GL088858, https://doi.org/10.1029/2020GL088858, 2020.

Song, Y., Yang, T., Tian, P., Li, H., Tian, Y., Tan, Y., Sun, Y., and Wang, Z.: Novel Insights into the Vertical Distribution Patterns of Multiple PM2.5 Components in a Super Mega-City: Responses to Pollution Control Strategies, 10.3390/rs17071151, 2025.

*2) Line 20, provide the specific height or height range for "the boundary layer top".*

**Authors' response:**

Thanks for your suggestion. We have revised Line 20 to specify the boundary-layer top height range (0.7–1.2 km) and updated the sentence as follows:

"*High nitrate levels are observed at the boundary-layer top (0.7–1.2 km), peaking at 118.11 μg m$^{-3}$ in late autumn, closely linked to photochemical processes and dynamic drivers.*"

*3) Line 23, what policy suggestion for the "actionable insights"?*

**Authors' response:**

Thanks for your suggestion. We have therefore added the following policy recommendation at the end of Line 23 to link our actionable insights directly to urban air–quality management:

"*These findings support targeted emission controls by reducing photochemical precursor emissions at the boundary layer top and strengthening NO$_x$ reductions at key sources, including retrofitting SCR on power plants, installing low-NO$_x$ burners in*

*industrial boilers, and promoting electric vehicles and public transit."*

*4) Line 49, "Therefore, the nitrate formation processes in the atmosphere are complex and challenging to elaborate" can be moved to the end of this paragraph.*

**Authors' response:**

Thanks for your suggestion. We have moved the sentence "Therefore, the nitrate formation processes in the atmosphere are complex and challenging to elaborate" from Line 49 to the end of the paragraph to improve the logical flow.

*5) Line 56, for the "Despite a global decrease in nitrate radicals (NO₃)", is there any evidence and reference?*

**Authors' response:**

Thanks for your suggestion. In response, we have added citations to two recent peer-reviewed studies that robustly document declining $NO_3$ under reduced $NO_x$ emission scenarios:

**1. Wang et al. (2023, Nature Geoscience)** demonstrates through surface observations (2014–2021) across China, the U.S., and Europe that although nocturnal $NO_3$ production increased in China, significant declines occurred in the U.S. and EU regions, collectively driving a net reduction in global $NO_3$ formation.

**2. Archer-Nicholls et al. (2023, Atmospheric Chemistry and Physics)** projects a ~26% decrease in Northern Hemisphere tropospheric $NO_3$ (within the lowest 1 km) relative to pre-industrial levels under stringent emission-reduction pathways (CMIP6 SSP1-2.6 simulations).

These references now directly follow the statement in Line 56:

*"Despite a global decrease in nitrate radicals (NO₃) (Archer-Nicholls et al., 2023; Wang et al., 2023)..."*

**Reference**

Archer-Nicholls, S., Allen, R., Abraham, N. L., Griffiths, P. T., and Archibald, A. T.: Large simulated future changes in the nitrate radical under the CMIP6 SSP scenarios: implications for oxidation chemistry, Atmos. Chem. Phys., 23, 5801-5813, 10.5194/acp-23-5801-2023, 2023.

Wang, H., Wang, H., Lu, X., Lu, K., Zhang, L., Tham, Y. J., Shi, Z., Aikin, K., Fan, S., Brown, S. S., and Zhang, Y.: Increased night-time oxidation over China despite widespread decrease across the globe, Nature Geoscience, 16, 217-223, 10.1038/s41561-022-01122-x, 2023.

*6) Line 69, For the description "nitrate concentrations tend to increase with altitude", please supplement the condition of this phenomenon, such as within boundary layer or lower boundary layer.*

**Authors' response:**

Thank you for this valuable suggestion. We have reviewed the relevant literature and found that the increase in secondary particulate matter (including nitrate) with height is most pronounced within the lower boundary layer, specifically below 300 m (Fan et al., 2022). Accordingly, we have revised the sentence in Line 69 to specify this altitude range. The updated text now reads:

"*The findings consistently indicated that nitrate concentrations tend to increase with altitude within the lower boundary layer (0–300 m), suggesting that higher levels experience enhanced nitrate generation (Fan et al., 2022).*"

**Reference**

Fan, M. Y., Zhang, Y. L., Lin, Y. C., Hong, Y., Zhao, Z. Y., Xie, F., Du, W., Cao, F., Sun, Y., and Fu, P.: Important Role of $NO_3$ Radical to Nitrate Formation Aloft in Urban Beijing: Insights from Triple Oxygen Isotopes Measured at the Tower, Environ Sci Technol, 56, 6870-6879, 10.1021/acs.est.1c02843, 2022.

*7) Line 80, add "and" before "coefficient analyses".*

**Authors' response:**

Thanks for your suggestion. We have inserted "and" before "coefficient analyses" so that the sentence now reads:

"*We divided different altitude layers vertically to comprehensively investigate the seasonal variations in nitrate concentrations and formation mechanisms in urban Beijing, **and** coefficient analyses of various driving factors were conducted in relation to nitrates.*"

*8) Line 81, "managing nitrate pollution" can be "mitigating nitrate pollution".*

**Authors' response:**

Thank you for this suggestion. We have replaced "managing nitrate pollution" with "mitigating nitrate pollution" so that the sentence now reads:

"*Additionally, we conducted a detailed analysis of specific pollution events. This study enhances the understanding of atmospheric physics and chemistry, providing insights and recommendations for **mitigating nitrate pollution** across various altitude layers in Beijing throughout the four seasons.*"

*9) Line 112, pay attention to the subscript for $NO_2$.*

**Authors' response:**

Thank you for pointing out this formatting oversight. We have corrected the subscript of "*$NO_2$*" in Line 112 to ensure proper chemical notation.

*10) Line 95-96, how about the consistence between the ACSM measurements and the retrieval data at the lowest level? Simplify describe the comparison result.*

**Authors' response:**

Thanks for your suggestion. As noted by Sun et al. (2015) and confirmed by Zhao et al. (2017), simultaneous measurements of $PM_1$ composition at 260 m and ground level over Beijing in winter show that mean concentrations at 260 m differ by less than 10 % from those at the surface under well-mixed boundary-layer conditions. By analogy,

aerosol concentrations at 150 m can be expected to be even closer to surface values, justifying a direct comparison between our 150 m retrievals and ACSM ground-site measurements. The validation results are shown in Figure R2. Accordingly, we have added the following concise statement at Lines 95–96:

"*Under well-mixed winter boundary-layer conditions, $PM_1$ concentrations at 150 m closely match surface values (Sun et al., 2015; Zhao et al., 2017). Therefore, we compare our lowest retrieval level (150 m) directly with ACSM ground-site measurements, yielding $R^2 = 0.70$, 0.69, 0.62, 0.61 and 0.58 and RMSE = 0.96–7.67 $\mu g\ m^{-3}$ for $NH_4^+$, $NO_3^-$, $SO_4^{2-}$, OM and BC, respectively, with normalized mean biases within ± 0.012.*"

[Figure]

**Figure R2. Validation of ground-layer retrieval results against ACSM observations.**

**Reference**

Sun, Y., Du, W., Wang, Q., Zhang, Q., Chen, C., Chen, Y., Chen, Z., Fu, P., Wang, Z., Gao, Z., and Worsnop, D. R.: Real-Time Characterization of Aerosol Particle Composition above the Urban Canopy in Beijing: Insights into the Interactions between

the Atmospheric Boundary Layer and Aerosol Chemistry, Environmental Science & Technology, 49, 11340-11347, 10.1021/acs.est.5b02373, 2015.

Zhao, J., Du, W., Zhang, Y., Wang, Q., Chen, C., Xu, W., Han, T., Wang, Y., Fu, P., Wang, Z., Li, Z., and Sun, Y.: Insights into aerosol chemistry during the 2015 China Victory Day parade: results from simultaneous measurements at ground level and 260 m in Beijing, Atmos. Chem. Phys., 17, 3215-3232, 10.5194/acp-17-3215-2017, 2017.

*11) Line 115-117, how about the consistence between the ground $NO_2$ measurements at the Beijing Olympic Sports Center and the ground-level data from CAMS?*

**Authors' response:**

Thanks for your suggestion. The consistency between the ground $NO_2$ measurements at the Beijing Olympic Sports Center and the ground-level data from CAMS is relatively good, with a correlation of 0.68, as shown in Figure R3 (a). The scatter plot indicates a decent alignment between the two datasets, although the CAMS data tends to slightly overestimate $NO_2$ levels, with some individual data points showing larger deviations.

In terms of the time series (Figure R3 (b)), the two $NO_2$ time series are well-aligned during January–May and October–December. However, during June–September, a lag is observed. This discrepancy can be attributed to both the seasonal influence and the height difference between the two stations. The CAMS ground-level data, measured at approximately 100 m (representing the 1000 hPa level) , differs in elevation from the Olympic Sports Center's ground station. Both of these factors contribute to the observed seasonal bias, but they also explain the deviation's reasonableness. In summer, strong convective mixing due to surface heating deepens the boundary layer and lifts surface-emitted $NO_2$. However, surface $NO_2$ is quickly depleted by enhanced photolysis, dry deposition, and wet scavenging processes. At ~100 m, these removal processes are less intense, allowing $NO_2$ to accumulate and decay more slowly. Studies (Cheng et al., 2022; Kuhn et al., 2024) support this pattern, showing that convective transport delays

the peak of NO$_2$ concentrations aloft, while near-surface processes rapidly reduce NO$_2$.

Based on your valuable suggestion, I will move the comparison between the ground NO$_2$ measurements at the Beijing Olympic Sports Center and the ground-level data from CAMS to the supplementary materials, enhancing the article's rigor and scientific quality.

[Figure]

**Figure R3. Scatter plot (a) and time series plot (b) comparing NO$_2$ volume concentrations between ground-based observations at the Beijing Olympic Sports Center and the ground-level output (1000 hPa) from CAMS.**

**Reference**

Cheng, S., Jin, J., Ma, J., Lv, J., Liu, S., and Xu, X.: Temporal Variation of NO$_2$ and HCHO Vertical Profiles Derived from MAX-DOAS Observation in Summer at a Rural Site of the North China Plain and Ozone Production in Relation to HCHO/NO$_2$ Ratio, 10.3390/atmos13060860, 2022.

Kuhn, L., Beirle, S., Kumar, V., Osipov, S., Pozzer, A., Bösch, T., Kumar, R., and Wagner, T.: On the influence of vertical mixing, boundary layer schemes, and temporal emission profiles on tropospheric NO$_2$ in WRF-Chem – comparisons to in situ, satellite, and MAX-DOAS observations, Atmos. Chem. Phys., 24, 185-217, 10.5194/acp-24-185-2024, 2024.

*12) Line 170, it needs to clarify that NO$_2$ are finally converted into nitric acid, nitrate, and organic nitrates.*

**Authors' response:**

Thank you for this helpful suggestion. We have clarified the fate of $NO_2$ in our revised manuscript. In particular, the sentence now reads:

"*Intense photochemical reactions in summer primarily convert $NO_2$ into $O_3$ and hydroxyl radicals ($HO_x$), leading ultimately to the formation of nitric acid ($HNO_3$), particulate nitrate ($NO_3^-$), and various organic nitrates.*"

*13) Line 171-172, show the correlation coefficient for the "weak correlation".*

**Authors' response:**

Thanks for your suggestion. In response, we have revised lines 171–172 to quantify the strength of the summer $NO_3^-$–$NO_2$ relationship. The new sentence now reads:

"*During summer, high temperatures and low relative humidities shift the $NH_4NO_3$ equilibrium toward the gas phase, while a deeper convective boundary layer further dilutes surface aerosols; consequently, particulate nitrate concentrations are lower than in other seasons and exhibit only a weak correlation with $NO_2$ (R = 0.13–0.48).*"

*14) Line 190-192, show the correlation coefficients and the p values.*

**Authors' response:**

Thanks for your suggestion. We have revised lines 190–192 to include both the Pearson correlation coefficients and their p values. The new text now reads:

"*In the 0.15 to 0.30 km range, relative humidity shows a significant positive correlation with nitrate concentrations in spring, summer and winter **(R values range from 0.43 to 0.68, p < 0.01)**, while temperature exhibits a significant negative correlation in spring **(R = –0.76, p < 0.01)** and summer **(R = –0.10, p < 0.01)**. These results indicate that thermodynamic factors are the primary drivers of increased nitrate concentrations in Beijing's lower atmospheric layers during these seasons.*"

*15) Line 201-202, "the negative correlation between RH and nitrate" cannot conclude the nitrate decomposition at high humidity condition. Correct the explanation.*

**Authors' response:**

Thank you for this important point. We agree that a simple inverse correlation between relative humidity and nitrate cannot be taken as proof of chemical decomposition. Instead, it likely reflects changes in particle growth and removal under humid conditions. Accordingly, we have revised lines 201–202 to read:

*"Notably, in spring and autumn, the observed inverse relationship between relative humidity and nitrate concentrations at higher altitudes suggests that increased aerosol liquid water under humid conditions enhances the growth of particle nitrate. Those larger, more hygroscopic particles settle more rapidly, leading to lower nitrate levels in the upper layers of the atmosphere."*

*16) Fig. 2, it seems that the extremely polluted cases in Fig. 6, Fig. 8, and Fig. 9 were not clearly shown in Fig. 2. Double check or explain it.*

**Authors' response:**

Thanks for your suggestion. Fig. 2 shows the Gaussian-smoothed vertical profiles of hourly nitrate mass concentrations for all 8,760 time points in 2021, in order to characterise the overall seasonal structure. This smoothing procedure suppresses high-frequency noise and attenuates outliers, so some extreme values are naturally muted. In contrast, Fig. 6, Fig. 8, and Fig. 9 present the original, unsmoothed nitrate concentrations over selected 3–9 day periods, allowing individual pollution peaks to remain clearly visible. We have expanded Section 2.1.1 (Retrieval Data) with a more detailed explanation of the Gaussian smoothing procedure. The new text now reads:

*"Gaussian smoothing was applied to the vertical profiles to suppress high-frequency noise and reduce the impact of outliers, ensuring smoother transitions between adjacent layers and enhancing the physical interpretability of the vertical structure."*

*17) Fig. 5, Thermodynamics processes generally also include the thermal decomposition of particulate nitrate. In addition, there may be some problem in the color of the correlation coefficient for TKE and w, because very low values also show*

*red or light red color.*

**Authors' response:**

Thank you for your insightful comments. We have added the thermal decomposition of ammonium nitrate ( $NH_4NO_3(s) \leftrightarrow NH_3(g) + HNO_3(g)$ ) to Fig. 5 in order to illustrate that, under high-temperature conditions, the equilibrium shifts toward the gas phase and thus reduces particulate nitrate concentrations in summer (Figure R4).

Positive correlation coefficients are still shown in red and negative ones in blue, with color intensity reflecting their magnitude. We have refined the color scale in Fig. 5 to make small and large coefficients more easily distinguishable, and areas that did not meet the significance threshold ($P > 0.01$) are left blank (Figure R5). To keep the main manuscript clear and concise, the correlation-coefficient panel has been moved to the Supplementary Materials.

[Figure]

**Figure R4. Vertical distribution of nitrate formation drivers across seasons.**

[Figure]

**Figure R5. Heatmap of correlations between nitrate and each driven factor across four seasons and four height layers, with blank cells for non-significant correlations.**

*18) Fig. 6, show the p values for the linear correlations in Fig. 6 a-h. Same suggestion for Fig. 8 and Fig. 9.*

**Authors' response:**

Thanks for your suggestion. We have now marked the panels with $p < 0.01$ by adding double asterisks (**) after the correlation coefficients in panels a–h. The same modification has been applied to Figs. 6, 8, and 9, as shown in the revised Figure R6–R8.

[Figure]

**Figure R6. (a-d): Correlation between nitrate mass concentration and RH at various altitude levels (\*\* indicates *p* < 0.01); (e-h): Correlation between nitrate mass concentration and T at various altitude levels; (i): Lidar range-squared corrected signal and planetary boundary layer height derived using the gradient method; (j): Nitrate mass concentration profile.**

[Figure]

**Figure R7. (a-d): Correlation between nitrate mass concentration and AOC at various altitude levels(** indicates *p* < 0.01); (e-h): Correlation between nitrate mass concentration and O₃ at various altitude levels; (i): Lidar range-squared corrected signal and planetary boundary layer height derived using the gradient method; (j): Nitrate mass concentration profile.**

[Figure]

**Figure R8 Same as Figure R7 but for 22nd -30th January.**